# Assessment of Fertility Dynamics and Nutritional Quality of Potato Tubers in a Compost-Amended Mars Regolith Simulant

**DOI:** 10.3390/plants13050747

**Published:** 2024-03-06

**Authors:** Antonio Giandonato Caporale, Roberta Paradiso, Mario Palladino, Nafiou Arouna, Luana Izzo, Alberto Ritieni, Stefania De Pascale, Paola Adamo

**Affiliations:** 1Department of Agricultural Sciences, University of Naples Federico II, Piazza Carlo di Borbone 1, 80055 Portici, Italy; ag.caporale@unina.it (A.G.C.); mpalladi@unina.it (M.P.); nafiou.arouna@unina.it (N.A.); depascal@unina.it (S.D.P.); adamo@unina.it (P.A.); 2Department of Farmacy, University of Naples Federico II, Via Domenico Montesano 49, 80131 Naples, Italy; luana.izzo@unina.it (L.I.);

**Keywords:** MMS-1, bioregenerative life support systems (BLSSs), in situ resource utilization (ISRU), *Solanum tuberosum* L., nutrient availability, tuber nutritional quality, antinutritional compounds

## Abstract

Mars exploration will foresee the design of bioregenerative life support systems (BLSSs), in which the use/recycle of in situ resources might allow the production of food crops. However, cultivation on the poorly-fertile Mars regolith will be very challenging. To pursue this goal, we grew potato (*Solanum tuberosum* L.) plants on the MMS-1 Mojave Mars regolith simulant, pure (R100) and mixed with green compost at 30% (R70C30), in a pot in a cold glasshouse with fertigation. For comparison purposes, we also grew plants on a fluvial sand, pure (S100) and amended with 30% of compost (S70C30), a volcanic soil (VS) and a red soil (RS). We studied the fertility dynamics in the substrates over time and the tuber nutritional quality. We investigated nutrient bioavailability and fertility indicators in the substrates and the quality of potato tubers. Plants completed the life cycle on R100 and produced scarce but nutritious tubers, despite many critical simulant properties. The compost supply enhanced the MMS-1 chemical/physical fertility and determined a higher tuber yield of better nutritional quality. This study demonstrated that a compost-amended Mars simulant could be a proper substrate to produce food crops in BLSSs, enabling it to provide similar ecosystem services of the studied terrestrial soils.

## 1. Introduction

In recent years, future long-term manned missions in deep space and the possibility to explore other planets of the solar system have been seriously considered by government space agencies and private commercial companies. However, the prolonged human permanence on orbital stations and planetary colonies will need specific technologies to regenerate essential resources, like air and water, and to exploit materials available in situ, while producing food and recycling waste [1].

In bioregenerative life support systems (BLSSs), selected organisms are combined on the basis of their metabolic paths in successive steps of crew waste recycling (feces, urine, carbon dioxide, and food residues) into oxygen, edible biomass, and potable water [2]. Accordingly, BLSSs for space will be realized through the integration of compartments hosting living organisms, integrated with physicochemical processes, to realize secure and reliable regeneration processes [3,4]. Moreover, the in situ resource utilization (ISRU), including the use of local soils, will contribute to achieve the self-sustenance of space colonies.

Edible plants are efficient bioregenerators, able to perform essential functions for human survival in extra-terrestrial environments, such as air renovation through photosynthesis, water purification through transpiration, and waste recovering through mineral uptake, while supplying fresh food and wellbeing to space crews [5]. Several crops, including fruit and leafy vegetables, cereals, and tuberous species, have been assessed as possible candidates for space cultivation, after considering specific constraints and technical and dietary requirements of the different mission scenarios. Agronomical features include compact plant size, a fast growing rate, elevated productivity and nutritional and nutraceutical value, and a high harvest index (HI), implying a limited volume of waste [6,7]. Among candidate crops, both potato (*Solanum tuberosum* L.) and sweet potato (*Ipomoea batatas* Poir.) are included as geophyte plants, producing edible underground organs.

Potato is highly productive, showing a good HI (0.7–0.8), offering several advantages, including a great number of genotypes with different features, and the staggered production of tubers, rich in carbohydrates and proteins and suitable for several food preparations [8]. However, it is typically grown in the field, and knowledge about plant physiology and productivity mainly refers to the outdoor farming on soil, while only a few studies concern the soilless cultivation in growth chambers. Particularly, the plant response to different hydroponic systems, nutrient solution composition, and environmental conditions was studied by the National Aeronautics and Space Administration (NASA) [8,9] and the European Space Agency (ESA) [10], in potato genotypes suitable for BLSSs. In addition, the success of the tuberization process has been verified in ground conditions in a modular system prototype for the cultivation of tuberous crops in microgravity [11].

Substrate is a crucial factor for potato cultivation since the habitability for the root system is pivotal not only for the proper plant growth but also for the optimal tuberization process. In general, non-arid loam and sandy-loam soils, with a low mechanical resistance to the tuber growth, and a pH interval of 5.5–8.0 (optimal value 6.0–6.5) are the most suitable [12]. However, in future plant cultivation on Mars, fertile soils could be unavailable and the in situ materials, including the Martian regolith and the organic waste of the mission, could be the only resources for assembling growing substrates [13].

In Mars-oriented research, since the real regolith cannot be used, investigations on Earth are carried out on commercial simulants, obtained from crushed terrestrial rocks, mimicking the geotechnical and physicochemical features observed in true regolith samples collected in robotic missions [5]. The MMS-1 Mojave Mars regolith simulant contains plagioclases, amorphous materials and zeolite, releasing essential nutrients (e.g., K, Ca, Mg, and Fe) for plant nutrition, but lacking those sourced from organic matter (organic C, N, P, and S) [14,15]. Obviously, the MMS-1 simulant does not represent the complexity of the entire surficial layer of Mars regolith, which shows high heterogeneity and spatial variability, analogously to Earth’s crust [5]; nevertheless, its chemical composition and mineralogical patterns is as much as similar to those of Mars regolith collected by rovers and robotic spacecrafts. Moreover, the MMS-1 simulant was found to be a coarse-textured and alkaline (pH 8.86) substrate, with scarce content of fine colloidal particles and low water-holding capacity. Consequently, to properly support the growth of plants, it requires an appropriate organic amendment (i.e., plant residues, human waste) to enhance the nutrient availability and the water retention, while contributing to dispose the organic effluents of the mission.

Until recent times, the use of Mars simulants amended with organic matter for plant cultivation in BLSSs was scarcely investigated [16,17,18]. In the last years, our team started a series of experiments focused on the characterization and exploitation of Mars simulants as plant growth media [5], also in mixture with organic materials of a different nature (i.e., green compost, peat, horse manure), mimicking the possible waste of a Mars mission (urine, feces, plant residues). In the first studies, lettuce was used as a model for leaf vegetables with a short cycle [14,15,19,20,21]. Later, experiments started also in tuber plants (potato) [22] and seed species (soybean). Overall, our results demonstrated that the organic amendment reduced the alkalinity and increased the nutrient availability, making the Mars regolith simulant suitable for plant growth.

In this experiment, we assessed the plant performance of potato plants grown in pots in an unheated glasshouse, on six substrates: the MMS-1 Mojave Mars simulant, pure and amended with a green compost (70:30 *v*:*v*), a fluvial sand, pure and mixed with compost at the same rate, a red soil, and a volcanic soil. We reported the results on the physicochemical properties of substrates before cultivation, and plant physiology and growth parameters in Caporale-Paradiso et al. [22]. In this paper, we show detailed data on the (i) nutrient bioavailability and chemical/physical fertility of the substrates after the plant growth cycle (to study their evolution from the starting point); and (ii) the elemental profile and nutritional quality of potato tubers yielded in the pot trial.

Our research questions were whether and to what extent the growth of potato plants and the production of tubers could modify the chemical features (i.e., content of essential nutrients, pH, EC) and the most relevant structural properties affecting the water-holding capacity (i.e., porosity) of regolith-based substrates could be reused in consecutive cultivations, specifically, whether the in-depth analysis of the evolution of growing media as an effect of plant cultivation provides information on the substrate fertility over time is useful in potato and, more in general, in long-cycle crop rotation. This knowledge is of crucial importance in the specific scenario, since relying on fertile substrates, able to sustain the plant growth in successive growing cycles, is a fundamental requirement to develop stable and reliable crop systems, suitable to fulfil vital regeneration functions (i.e., air renovation and water purification) and to produce food in a predictable and durable way. In addition, the insight of the influence of different substrates on the plant food quality (i.e., potato tubers) is decisive to define balanced diets with proper nutrient intakes (based on the actual concentration in the plant product) and to ensure the food security in terms of anti-nutritional compounds (e.g., potato glycoalkaloids) to guarantee the astronauts’ survival in space, as well as to exploit nutraceutical properties of fresh food as a countermeasure to human diseases related to space factors acting as stressors on the human body (e.g., antioxidants and other health-promoting compounds).

## 2. Results

### 2.1. Physico-Hydraulic and Chemical Indicators of Substrate Fertility

The organic C, total N and S concentrations, and the C/N ratio in different growth media (separated in RH and BK soils) are shown in Table 1. The supply of these nutrients through the compost significantly raised their concentrations in R70C30 and S70C30 treatments, in comparison to terrestrial soils (VS and RS) and non-amended substrates (R100 and S100). No relevant variations were observed in the other substrates; as well, no statistically different concentrations of organic C, total N and S, were found between RH and BK soils (Table 1).

The bioavailable fractions of the main macro- and micronutrients, extracted from the different substrates after the plant growing cycle, are provided in Table 2 (readily soluble fractions) and Table 3 (potentially bioavailable pools). Despite the equal supply of nutrient solution to all the treatments, we found statistically significant differences in the nutrient availability among the substrates. For the majority of the elements (except K and Na), the readily soluble nutrient fractions (extracted by NH_4_NO_3_, a salt with weak acid hydrolysis) were lower than the potentially bioavailable fractions (extracted by EDTA, a complexing reagent) (Table 2 and Table 3), especially for micronutrients (values one or two orders of magnitude different). Additionally, we found an overall higher bioavailability of nutrients in the MMS-1-based substrates (often similar to terrestrial soils) than sand-based ones (Table 2 and Table 3); this was also true for readily soluble Ca, not for potentially bioavailable Ca extracted by EDTA capable of chelating the Ca present in the carbonates (largely occurring in fluvial sand; Table 1 in Caporale-Paradiso et al. [22]). Regarding the terrestrial soil, RS basically showed a higher bioavailability of nutrients than VS, and this was mainly due to the significantly higher clay content and CEC ([22]: Table 1).

Interestingly, the bioavailability of important plant nutrients (i.e., K, Mg, and Zn) was significantly higher in the RH than BK soils, which may be due to rhizodepositions, intense microbial activity, and rhizosphere pH acidification (Figure 1). On the other hand, a significant depletion of bioavailable B in the RH vs. BK soils was found, which may be due to a high nutrient uptake rate in the final plant growth period. The interaction between the factors of soil (S) × RH vs. BK (RB) was significant (*p* < 0.05) for K, Na, and B (Table 2 and Table 3), Mg and Mn (Table 2), and Fe (Table 3).

We also measured the pH (Figure 1A) and EC (Figure 1B) of the substrates after plant growth, two factors having a key role in nutrient availability. We found a significant depletion of pH values in the RH vs. BK soils (8.01 vs. 8.19, on average; Figure 1A) mainly due to rhizodeposition and enhanced microbial activity. This trend was particularly evident in the pure Mars simulant (R100) and sand (S100). In the neutral to sub-alkaline terrestrial soils, an opposite behavior was assessed, statistically significant only in the RS clay soil likely containing lots of fine short-range-ordered minerals. The amendment of simulant or sand with compost significantly mitigated their alkalinity (Figure 1A). Regarding the EC, we measured significantly higher values in S70C30, R70C30, and R100-RH than other media; this can be due to either the high content of organic matter or zeolite (in MMS-1, as assessed by Caporale et al. [14]), capable of retaining/releasing many ions on/from their own exchange surfaces. Unlike the pH (Figure 1A), the EC values in the RH soil were not statistically different from those of the BK soil (Figure 1B). Nevertheless, the interaction between the factors S × RB was significant both for the pH and EC (Figure 1A,B).

The bioavailability of P was assessed by the Olsen method (Figure 2), a proper and efficient procedure for neutral to alkaline soils. The supply of nutrient solution raised the availability of P in non-amended substrates (S100 and R100) in comparison to the starting point. The large occurrence of organic matter in S70C30 and R70C30 assured a high bioavailability of P in two substrates, significantly higher than two terrestrial soils. Indeed, the availability of P in the VS was higher than the RS, rich in Fe oxides owing a high phosphate adsorption capacity [23]. We also observed a slight, but not significant (Figure 2), increase in the available P in the RH soil, in comparison with the BK soil (except in R70C30), probably due to a possible combined effect of mycorrhizal fungi and P-solubilizing bacteria in the highly dynamic RH environment. The interaction between the factors S × RB was also not statistically significant (Figure 2).

A wide description of physical-hydraulic properties and retention curves of the six mixtures is available in Caporale et al. [22]. In this study, we provide the pore area distribution at different values of suction in Figure 3, arising from the numerical derivation of the retention curves. The sandy soil (S100) is found at one extreme, with a very sharp peak at very low suction values (around 20 cm). In contrast, the red soil (RS) and volcanic soil (VS) exhibit a well-graded characteristic distribution, indicating a better water retention in a wide range of suction values. It is noteworthy that the native regolith (R100) shows a pore frequency distribution that closely follows that of the volcanic soil (VS).

In Figure 3, we also indicated with two vertical dashed lines the suction values within which the root water uptake activity for the potato crop is most facilitated. In Figure 3, we also indicated with two vertical dashed lines the suction values within which root water uptake activity for potato is facilitated. We already highlighted the beneficial effect of compost supply on the total easily extractable water (TAW) for potato [22]. Here, we prove the increase of 125% in TAW potato for fluvial sand (from S100 to S70C30) and 26% for regolith simulant (from R100 to R70C30). Compost addition mainly increased the pores but decreased their frequency for suction values below 25 cm, both for pure sand and regolith.

### 2.2. Plant Growth

Detailed results about plant growth and physiology of potato cultivar ‘Colomba’ grown in the different substrates are reported in Caporale-Paradiso et al. [22]. For the sake of completeness, we summarize here the plant response in terms of biomass accumulation in the aboveground part and the underground organs (roots, stolons, and tubers).

At the end of the experiment (99 DAS), plants were significantly taller on fluvial sand (39.5 cm) and MMS-1 pure and amended with green compost (38.8 cm on average), while they were shortest on pure regolith (31.9 cm). The plant leaf area was greater in VS plants (1142.4 cm^2^ plant^−1^) and both the compost mixtures (915.7 cm^2^ plant^−1^ in R70C30 and 869.2 cm^2^ plant^−1^ in S70C30). Consistently, the dry matter accumulation in the epigeal part was higher on VS (3.28 g plant^−1^), followed by the mixtures (2.80 g plant^−1^ on average). The regolith simulant alone (R100) reduced the leaf development (526.0 cm^2^ plant^−1^) and the aerial biomass (1.87 g plant^−1^) compared to the other substrates.

Tuberization started 28 days after planting on average. Data on the hypogenous organs confirmed a greater dry weight on VS (18.5 g plant^−1^) and S70C30 (17.4 g plant^−1^), and also on S100 (18.6 g plant^−1^). The tubers yield was higher in the sandy substrates (101.7 g plant^−1^ on average), followed by VS and R70C30 (89.7 g plant^−1^ on average), and RS (75.0 g plant^−1^). Plants on R100 were the least productive (46.4 g plant^−1^). The tuber dry matter percentage was 19.2% on average in VS and RS, and 17.4% on average in the other plant growth media. Dry matter partitioning unveiled the highest HI in both the sandy media (>84%) and the lowest one in the regolith-based media (78.4%).

### 2.3. Tuber Quality

The multielement profile of potato tubers grown in the different substrates is provided in Table 4 (nutrient concentrations) and Table 5 (nutrient contents, obtained multiplying nutrient concentrations by dried tuber biomass).

Potato tubers contain high concentrations of C (and consequently of carbohydrates) and important nutrients such as K (on average, 21.6 g kg^−1^ DW) and N (13.5 g kg^−1^ DW); they also hold good concentrations of S (3.0 g kg^−1^ DW), P (2.0 g kg^−1^ DW), Mg (1.4 g kg^−1^ DW), Ca (0.3 g kg^−1^ DW), and other oligo-elements (i.e., Na, Fe, Zn, B, Mn, and Cu) present in concentrations lower than 0.1 g kg^−1^ DW (Table 4). Except for C and Ca, nutrient concentrations in tubers harvested from the different substrates were statistically different; basically, the potatoes grown in RS, S70C30, R100, and R70C30 showed a significantly better nutritional status than those in VS and S100 (Table 4). These findings agreed with the trends observed on the nutrient bioavailability in the substrates (readily soluble fractions in particular; Table 2) and are also related to the tuber biomass yielded by each treatment (Table 5 in Caporale-Paradiso et al. [22]). In other words, the low tuber biomass produced on R100 and RS accumulated higher concentrations of nutrients in their tissues in comparison with other substrate treatments producing a higher tuber biomass (e.g., VS and S100). This implies that non-amended simulant produced a scarce tuber yield, but of a high nutritional quality. However, it is noteworthy as the addition of compost to MMS-1 simulant led to the best agronomic outcome (i.e., the combination of a high yield with the best nutritional quality).

The data on the nutrient contents shown in Table 5 (i.e., the amount of nutrients in the tubers of each plant) highlighted once again that the compost addition to the poorly fertile substrates (R100 and S100) significantly raised the amounts of nutrients accumulated in the tuber biomass of the R70C30 (+100% on average, for all the nutrients) and S70C30 (+20%) plants, to values similar to the terrestrial plants (VS, in particular). For instance, the tubers produced by a R70C30 plant could theoretically provide an astronaut with a total of 753 mg of essential nutrients (i.e., K, N, S, P, Mg, Ca, and Na; value obtained by the sum of their contents), and 1124 µg of other healthy nutrients, such as Fe, Zn, B, Mn, and Cu (value obtained by the sum of their contents; Table 5). This is a good amount of nutrients, basically better than that theoretically available from potatoes yielded from the other substrates.

Data on the main quality parameters of tubers are summarized in Table 6. Tubers from S70C30 showed the highest total protein content, followed by those from R70C30, R100, and VS, and RS and S100, which gave the lowest value. The starch content was higher in tubers grown on terrestrial soils (VS and RS), and decreased progressively in those on S100, R100, S70C30, and in R70C30. The amount of total dietary fiber reached the highest content in R100 and the lowest in RS. The content of ascorbic acid did not differ in the different substrates. The α-chaconine was predominant compared to α-solanine in tubers from all the substrates. The highest contents of α-solanine, α-chaconine, and total glycoalkaloids (as the sum of α-solanine and α-chaconine) were found in VS and R100, whereas no significant differences were detected among those from all other substrates except for tubers from S100, which showed the lowest content of α-solanine.

## 3. Discussion

To grow properly and complete a normal life cycle, edible plants need essential nutrients sourced by soils or growth media, except for carbon, oxygen, and hydrogen which are obtained from air and water. Previous characterization studies of MMS-1 Mars simulant [14,20] demonstrated that the total content of many vital elements in the simulant could be adequate to fulfil the plant requirements. However, plants commonly absorb only the bioavailable fractions of mineral nutrients (i.e., the readily soluble and exchangeable forms), while they cannot use the elements integrated in mineral lattices, released only after mineral (bio)weathering [5]. Nutrient bioavailability and plant nutrition in the rhizosphere soil are governed by many dynamic processes and the pseudo-equilibrium between the water and the solid phases, rather than by the total concentrations of mineral nutrients. Several properties, including texture, clay content, pH, and EC, have a pivotal role in the regulation of the nutrient bioavailability and the root uptake [5]. Hence, the extraction and the quantification of bioavailable nutrient pools, matched with the measurement of soil pH and EC, are fundamental to understand the (bio)chemical rhizosphere processes, strongly affecting the nutrient uptake and the plant nutritional status.

In this study, the bioavailability of the nutrients was assessed by chemical extractants which simulate the solutions circulating in the rhizosphere environment. The single-step 1 M NH_4_NO_3_ soil extraction is widely used to quantify the readily soluble and easily bioavailable fractions of the elements in the soil system [24]. Soil extraction with 0.05 M EDTA at pH 7, instead, allows to quantify the potentially bioavailable pools of elements in the soil, since EDTA can chelate several metal ions [25] and can partially extract metals organically bound or occluded in secondary minerals and oxides [26]. Our extractions demonstrated that all the substrates provided bioavailable fractions of essential nutrients to potato plants (Table 2 and Table 3); however, when not amended with compost, both Mars simulant and fluvial sand cannot fulfil the plant requirements if not adequately supported by fertigation. This aspect was also addressed in a growth experiment on Mars and Lunar simulants amended with a horse and swine manure [20], through a comparison between the nutrient requirements of lettuce plants and the bioavailable nutrients extracted by extraterrestrial simulants. The lower bioavailability of nutrients in non-amended substrates had an evident negative impact on potato plant physiology and productivity [22]. Moreover, the significant reduction of pH values in the RH vs. BK soils (Figure 1A) also evidenced a greater effort (likely, exudation of acid organic compounds) by the plant to mobilize vital nutrients from these poorly fertile soils.

The addition of stable organic matter to mineral substrates made them more similar to the terrestrial soils, where humified organic matter and related microbiota interact with the mineral moiety to form a porous and highly dynamic environment. Martian regolith, and its terrestrial simulants, in fact, lack the essential microbial communities commonly present in the organic matter of the terrestrial soils, which have a key role in the rhizosphere nutrient cycling and plant growth processes [5]. Thus, when mixed with a quality compost, a Martian regolith simulant such as MMS-1 can be able to provide similar ecosystem services to terrestrial soils. Accordingly, the overall nutrient bioavailability in the amended substrates reached the same order of magnitude of VS and RS terrestrial soils (Table 2 and Table 3, Figure 2). Hence, the addition of composted organic material to the alkaline and poorly fertile R100 and S100 basically enhanced their (bio)chemical fertility and lowered the alkaline pHs. Additionally, the growth of potato roots and tubers and the intense release of exudates increased the pool of organic C and total N in the S100 and R100 substrates (Table 1), if compared with the starting point ([22]: Table 1); indeed, the supply of nutrient solution contributed to enriching the two substrates of N as well. The transformation of fresh organic material into high-quality compost to recycle as crop amendment is a key step to assure a sustainable use of scarce resources in extraterrestrial BLSSs [5].

The primary constraints of regolith are the limited availability of nutrients for biological processes and the inadequate ability to retain water, which is attributed to the absence of organic carbon and fine-grained colloidal particles [27,28]. Adding compost to regolith can improve several properties, including water-holding capacity, medium structure, porosity, and permeability. We quantified how the compost improved the water retention capacity of the regolith within the physiological limits suitable for potato root water uptake. Furthermore, compost can improve soil and regolith structure by promoting aggregation of soil particles and increasing porosity, allowing for a better water infiltration and gas exchange between the solid phase and the atmosphere.

The analysis of the pore area distribution (Figure 3) revealed distinct water retention characteristics among the different soil types, with sandy soil (S100) showing a sharp peak at low suction values, while red soil (RS) and volcanic soil (VS) showed a well-graded distribution, indicating better water retention over a wider range of intake values. The addition of compost to both sandy soil and regolith demonstrated significant improvements in total readily extractable water (TAW) and pore frequency distribution. The physical properties of MMS-1, including particle size distribution, particle density, and bulk density, influenced their overall porosity and saturated water content [20]. This suggests that the compost improves the water-holding capacity, particularly for suction values below 25 cm. This finding, in addition to its intrinsic value, allows for more efficient irrigation management from an energy point of view, since the irrigation objective can be achieved with a reduced number of irrigation events.

The average temperatures occurred in greenhouse throughout the experiment were greater than the optimal level for potato for tuber sprouting (15 °C), and close to the optimal values during tuberization (18 °C) [29].

The ‘Colomba’ potato plants showed a good growth performance in pots in the fall–winter period, in the unheated glasshouse in a Mediterranean climate, and all the studied plant growth media (MMS-1 simulant as well) allowed the development of a good amount of biomass and the process of tuberization. This confirms the good adaptability to different root environments known for the crop, as observed in previous space-oriented experiments in phytotron comparing several possible substrates and containers, such as a peat-based mixture in cylindrical boxes [30], a peat–vermiculite mix in trunk conical pots [8], a cellulosic sponge in rectangular trays [11], and in different cultivation systems, including the nutrient solution only (NFT) with Molders et al. [10].

The growth of the plant epigeal part was promoted by volcanic soil and both sand and Mars regolith simulant mixed with green compost, compared to the same non-amended substrates, confirming the need for soil organic matter for potato cultivation [31] (p. 928). Accordingly, the development of the hypogenous organs (i.e., roots, stolons, and tubers) was considerable on the same substrates, and also on pure sand. On the other hand, the Mars regolith simulant MMS-1 was unable to properly support the plant growth, presumably because of the scarce water retention, the high pH, and the scarcity of organic carbon and essential mineral nutrients. However, the addition of compost improved the physical and chemical properties (structure, water, and nutrient availability) and the overall fertility of the regolith, with positive effects on the crop productivity, as previously observed in lettuce on the same media [15].

Potato plants on regolith simulant developed the shortest roots and the greatest percentage of thicker roots, hence a root system impairing the plant resource acquisition [32]. In fact, the finer roots increase the root surface, as well as the explored soil volume, and usually allow a higher nutrient uptake capacity per unit of root mass [33]. In plants on the regolith simulant, the reduced percentage of fine roots and the higher proportion of ticker roots could indicate a possible mechanism of resource conservation [34].

Our plants completed the tuber-to-tuber cycle and produced healthy tubers on all the studied substrates, in a predictable time for the Colomba potato cultivar. This result agrees with that obtained in ‘Colomba’ in the growth chamber from pre-sprouted tuber seeds, on both a peat-based mixture [30] and a cellulosic sponge [11]. However, in this experiment, the final tuber yield was lower than that expected for mini-tuber potato plants (data provided by the breeder for the cultivar; www.hzpc.com; accessed on 8 January 2024), while the percentage of dry matter was in line with the reference value for the cultivar. The potato productive performance depends on several factors, including genotype, climate, and soil features [31] (p. 928). In this experiment, temperatures experienced by plants during the early developmental stages, together with the relatively low solar radiation due to the greenhouse frame and cover, and the cloudy weather of the period might have limited the growth rate and anticipated the senescence. Nevertheless, the tuber number is indicative of the potential plant productivity, as the tubers developed in the initial 3 weeks of tuberization will determine the majority of the final yield [31].

An adequate intake of mineral nutrients is crucial for the health of astronauts, to meet their nutrient needs and to counteract the detrimental effects of the space environment [35]. According to the multielement profile of tubers (Table 4 and Table 5), the consumption of potatoes (adequately cooked) can allow space crews to intake minerals such as K, N (primarily), S, P, Mg, and Ca (secondly), and essential micronutrients (i.e., Na, Fe, Zn, B, Mn, and Cu). A sufficient intake of minerals is crucial for sustaining the astronaut’s wellbeing. Potassium, for instance, abundantly sourced by potato tubers, helps to keep normal fluid levels in human cells (while Na, its counterpart, maintains normal fluid levels outside of cells) and normal blood pressure, and helps with muscle contraction [36]. It is noteworthy as the nutrient concentrations in potato tubers grown on Mars simulant, sole or amended with compost, are similar to that of tubers grown in our terrestrial soils from Italy (i.e., VS and RS), as well as in Brazilian [37], Canadian [38]*,* and Chinese [39] soil environments. The amendment of MMS-1 simulant with compost improved both the productivity of potato plants [22] and the nutritional value of tubers (in terms of concentrations and contents of healthy elements; Table 4 and Table 5). The compost applied as a soil amendment was found to increase the potato tuber yield and size/quality in many studies [40,41]. These increases can be attributed to both ‘nutrient’ (i.e., slow-release source of elements) and ‘non-nutrient’ (e.g., increase in soil water retention) benefits related with the organic matter [42].

Potatoes are rich in a variety of nutrients, including proteins, carbs, vitamins, dietary fibers, minerals, and a number of other health-promoting compounds, and also contain a few compounds with harmful effects, if ingested in excess [43] (pp. 191–211). To ensure safe human consumption, the content of 20 mg/100 g fw is recommended as the upper limit for glycoalkaloid content in potato [44]. The prevalent alkaloids in the Solanaceae family (and especially in the Solanum genus), synthesizing a variety of alkaloidal chemicals, are the glycoalkaloids, nitrogen-containing steroidal glycosides. Glycoalkaloids are found in tubers in variable quantities and may represent a source of water and soil contamination [45]. Their concentrations are influenced by several factors, including geographic location, genotypes and varieties, maturity at harvest, and growth conditions [46]. In Solanum genus, more than 80 glycoalkaloids were identified, although α-solanine and α-chaconine represent the most abundant [47]. The tuber content of glycoalkaloids obtained in our experiment is in line with previous studies in potato, which report values ranging from 0.9 to 37 mg/100 g for α-chaconine and from 0.4 to 17 mg/100 g for α-solanine [48,49,50].

## 4. Materials and Methods

### 4.1. Plant Cultivation

The experiment was carried out in an unheated glasshouse at the experimental facilities of the Department of Agricultural Sciences of the University of Naples Federico II (Portici, Italy—40°49′ N, 14°20′ E).

Plants of potato (*Solanum tuberosum* L.) cv. ‘Colomba’ (HZPC Holland B.V.) were grown in pots containing 4 L of substrate. The same cultivar, identified as suitable for cultivation in BLSSs, was previously evaluated for the response to different light spectra [30] and growing media [11], in a controlled environment. The experiment lasted from 27 September 2021 to 4 January 2022.

Six substrates were compared: the Mojave Mars regolith simulant MMS-1, pure (R100) and in mixture with a commercial green compost 70:30 *v*:*v* (R70C30), a fluvial quartz sand, pure (S100) and with compost at the same rate (S70C30), a red clay soil (RS) from Sicily (Italy), and a volcanic sandy loam soil (VS) from Campania (Italy). The mature green compost (Vivai Gardea, Verona, Italy) originated from pruning residual and grass swathe, under controlled conditions, in approximatively 3 months. A previous characterization revealed a low C/N ratio, denoting a good availability of N, and a significant aromatic component entailing a good organic matter stability [14]. It is worth noting that the high pH (8.25) did not allow to adjust the pH of the MMS-1/compost mixtures [14] at the level required to maximize the availability of nutrients.

Irrigation was driven by measurements of water potential in the tested substrates, assessed with tensiometers (Jet-filled 2725ARL—18”, Soilmoisture Equipment Corp., Goleta, CA, USA). Water availability was kept at the optimal level for potatoes (range of matric potential 25-320 cm; [51]), which was restored every time the measure revealed a value below the lower limit. Plants were fertigated with a complete nutrient solution, using the recipe for potato developed by Molders et al. [10], with pH 5.7 and electrical conductivity (EC) 1.68 dS m^−1^. Every 3 fertigations, one irrigation with only water was supplied to avoid the increase in salinity in the media. From the 4th week of cultivation, the N supply was suspended to accelerate the beginning of tuberization.

Air temperature and relative humidity at the canopy level were recorded every 10 min (wireless data logger Bluetooth RoHS). The day/night temperatures (mean ± standard deviation) in the different phenological phases were 30.0 ± 6.2/22.0 ± 2.7 °C (emergence), 22.8 ± 5.4/17.00 ± 2.5 °C (vegetative phase), and 16.2 ± 5.9/10.4 ± 3.0 °C (tuberization). The natural photoperiod decreased from 13 h and 00 min (27 September) to 9 h and 24 min (4 January), and the global solar radiation ranged from 7.12 MJ m^−2^ d^−1^ (1 October) to 0.55 MJ m^−2^ d^−1^ (29 November). The ambient CO_2_ concentration recorder in the daytime was approximately 430 ppm.

### 4.2. Sampling and Measurements

#### 4.2.1. Characterization of the Substrates

##### Physical and Hydrological Properties

The physical and hydrological properties were studied on the reconstructed samples and placed in steel cylinders (height 12 cm, diameter 8.5 cm). In addition to the usual properties shown in Caporale-Paradiso et al. [22], specific transient evaporation tests were carried out to obtain the water retention curves. The suction value at three levels and the mean water content were measured. The data for the retention curves were determined using the WIND technique and interpolated with the parametric relationship of van Genuchten [52]. The curves were then derived to obtain the frequency distribution of the total pore area for every suction value. The analyses allowed the study of both the differences between the native substrates and the effect of the compost addition to the MMS-1 regolith and fluvial sand.

##### Chemical Properties

During the sampling of the plants (99 days after sowing), potato tubers were harvested, and the root biomass was separated from the growth medium. The soil adhering to tubers and roots (tuberosphere/rhizo soil, RH) was separated from the bulk soil (BK). The RH and BK soil samples were air dried and sieved at 2 mm for the successive physicochemical analyses.

The pH was measured by a pH meter (Hanna Instruments 210) in ultrapure water, at the solid solution ratio (SSR) of 1:2.5, and EC by a conductivity meter (COND 70 + XS) at a SSR of 1:5.

The total C, N, and S contents were determined by a Micro Elemental Analyser (UNICUBE^®^, Elementar, Hesse, Germany). Calibration was carried out using a sulphanilamide standard (Elementar, 99.5%). A separate measure of carbonates was performed to distinguish the organic C from the total C content.

The readily soluble and potentially bioavailable fractions of the macro- and micronutrients were extracted from the substrates, at the end of the cultivation cycle, by 1 M NH_4_NO_3_ (solid/solution ratio: 1/2.5; reaction time: 2 h; ISO 19730:2008 [53]) and 0.05 M EDTA at pH 7 (solid/solution ratio: 1/10; reaction time: 1 h; [54]), respectively. The extracts were filtered in filter papers (Whatman 42) and analyzed by Inductively Coupled Plasma–Mass Spectrometry (ICP-MS, Thermo Scientific iCAP Q, Waltham, MA, USA). The certified reference material BCR 700 was employed to assess the quality of EDTA extractions (recovery at ±10% of the certified values).

The available fraction of P was solubilized in 0.5M NaHCO_3_ (buffered at pH 8.5) and determined by the colorimetric Olsen method.

#### 4.2.2. Plant Growth and Tuber Yield

Plant leaf area of fully grown plants (72 DAS) was estimated in 3 plants per treatment, through the analysis of digital images of all the plant leaves (non-destructive survey), using the ImageJ software 1.53 k (Wayne Rasband National Institute of Health, Bethesda, MD, USA).

The fresh weight and the dry weight after oven drying at 80 °C of the different plant portions (leaves, stems, roots, and tubers) were determined at the harvest (99 DAS), using an analytical scale (Gibertini Europe 500).

#### 4.2.3. Nutritional Quality Assessment of Potato Tubers

Fresh harvested tubers from the different substrates were washed, unpeeled, cut, immediately frozen at –80 °C, then freeze dried. The lyophilized material was ground to a fine powder, then stored at –20 °C.

##### Elemental Analysis

The concentrations of C, N, and S were measured in 2 mg of dried samples by the Micro Elemental Analyser—UNICUBE^®^ (Elementar, Hesse, Germany); the content of K, P, Mg, Ca, Na, Fe, Zn, B, Mn, and Cu was determined by ICP-MS (Thermo Scientific iCAP Q, Waltham, MA, USA), after digestion of 500 mg dried samples in a microwave system (Milestone Start D, Sorisole, BG, Italy) with HNO3 65% and HCl 37%.

##### Chemicals and Reagents

Methanol, acetonitrile, and formic acid (Merck, Darmstadt, Germany), and standards of α-Solanine and α-Chaconine (Sigma Aldrich, Milan, Italy), all of analytical grade, were used.

Starch and vitamin C contents were assessed by acid hydrolysis (AOAC 925.38) and indophenol (AOAC 967.21) methods, respectively [55]. The protein content was quantified through the Bradford protocol, using bovine serum albumin as the standard [56]. Total dietary fiber was determined by enzymatic-gravimetric assay using Megazyme Kit (K-TDFR) following the procedure reported in the instruction manual [57].

##### Glycoalkaloids Extraction

Glycoalkaloids were extracted following the procedure described by Maldonado et al. [58] with slight adjustments. In brief, 20 mL of 2% methanol acetic acid solution was added to 250 mg of lyophilized sample, stirred for 30 min, and centrifuged at 4000× *g* for 10 min. After that, the supernatant was collected and filtered through a 0.2-micron nylon filter. The analysis was performed by UHPLC-HRMS Orbitrap using α-Solanine and α-Chaconine as standard for calibration curves.

##### Liquid Chromatography Q-Orbitrap Mass Spectrometry Analysis

The analysis of Glycoalkaloid was performed by Ultra High-Pressure Liquid Chromatography (UHPLC, Dionex UltiMate 3000, Thermo Fisher Scientific, Waltham, MA, USA). Chromatographic separation was conducted through the use of a thermostated Gemini 3 µm C18 column (50 × 2 mm, Phenomenex, Torrance, CA, USA). The mobile phases were 0.1% formic acid in water (A) and acetonitrile with 0.1% formic acid. The gradient started with 15% B, a 5 min increase to 31% B, a rise to 35% B in 0.1 min, and in 0.9 min increase to 55% B. For 0.5 min, the gradient was set at 55% B, then lowered to 15% B for 0.5 min, and kept at 15% for 2 min. Two scan events were included in the MS method: full ion MS and all ion fragmentation (AIF). In AIF mode, the following parameters were set: scan range: 100–1000 m/z, mass resolving power: 17,500 FWHM, ACG target: 1 × 105. In full MS mode, the automatic gain control (AGC) target was set at 1×106 with a resolution power of 35,000 Full Width at Half Maximum (*m*/*z* 200)s. The ion source characteristics used were spray voltage 3.0 kV, capillary temperature 275 °C, S-lens RF level 50, heater temperature 350 °C for auxiliary gas, sheath gas pressure 35, auxiliary gas 15. The collision energy (CE) varied in the range between 10 and 60 eV. Identification and confirmation were performed at a mass tolerance of 5 ppm. Data analysis was performed using Xcalibur software, v. 3.1.66.10 (Thermo Fisher Scientific, Waltham, MA, USA).

### 4.3. Statistical Analysis

Each treatment consisted of 5 replicates (plants), randomly selected. Data were analyzed by one-way (nutritional and antinutritional compounds in tubers) or two-way (nutrient bioavailability and chemical fertility indicators in the substrates) analysis of variance (ANOVA), through the software IBM SPSS Statistics v27 (SPSS Inc., Chicago, IL, USA), with the following sources of variance: (i) six different soils/substrates (S; one-way and two-way ANOVA); two different soil types (RB; i.e., RH vs. BK; two-way ANOVA). Means were compared through the Duncan’s multiple range test (DMRT), at *p* < 0.05.

## 5. Conclusions

This experiment was carried out within a wider research program aiming at identifying the best substrate while exploiting in situ resources (i.e., Martian regolith and organic waste of the mission), to realize reliable and sustainable cultivation systems for candidate crops in planetary colonies.

Our analyses showed that the Mojave Mars regolith simulant MMS-1 lacks essential plant nutrients normally generated and geo-chemically regulated by organic matter (i.e., N, P, and S), and presents several features hampering the plant growth (e.g., high pH and Na content, poor physical structure, and low water-holding capacity). Consequently, the growth of potato plants on the regolith simulant alone limited both the epigeal and hypogeal growth compared to the other substrates, even in the presence of fertigation. The low tuber biomass obtained on pure regolith simulant determined a higher content of nutrients in their tissues (concentration effect) in comparison to other substrates, implying that MMS-1 produced a scarce tuber yield but with high nutritional quality. Nevertheless, the amendment with green compost improved the structure and general fertility of the medium, enhancing the plant performance, the overall dry matter accumulation, and the tuber yield and quality. The addition of this source of organic matter led to the best agronomic outcome, combining a high yield with the best tuber quality, and allowed the maintenance of a sufficient level of substrate fertility for successive cultivations.

In conclusion, compost amendment is a successful strategy to create long-lasting fertile substrates from the poor Mars regolith and the organic waste of the mission (here mimicked by green compost). This evidence represents useful information on the performance of potato (as a model of tuberous crop) in containers under protected cultivation and on plant response to the growth medium, which could contribute to develop efficient cultivation systems for resource bioregeneration in future Mars settlements. In this view, further investigations are in progress on the regolith-based substrates retrieved after the potato growing cycle used for cultivation of other candidate crops, including plant species typically improving soil fertility (i.e., Leguminosae performing atmospheric N-fixation).

## Figures and Tables

**Figure 1 plants-13-00747-f001:**
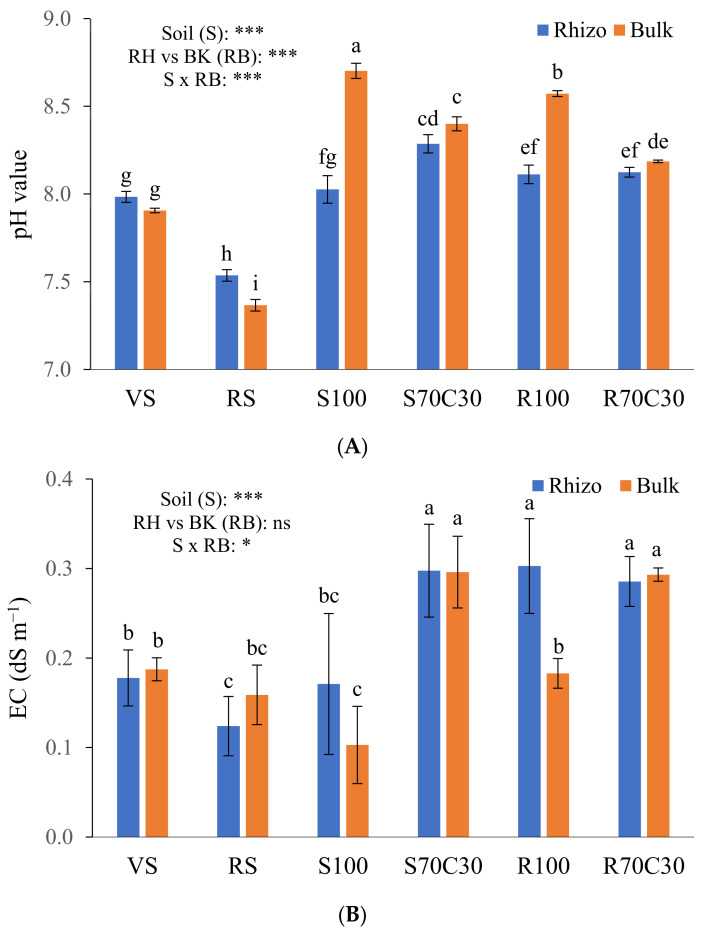
pH values (**A**) and electrical conductivity (EC) (**B**) of volcanic soil (VS), red soil (RS), fluvial sand, pure (S100) and mixed with green compost (70:30 *v*:*v*; S70C30), and Mojave Mars regolith simulant MMS-1, pure (R100) and amended with green compost (70:30 *v*:*v*; R70C30), separated (after potato plant growth) in potato tuberosphere/rhizo (RH) and bulk (BK) soils. Bars indicate mean values of 5 replicates ± standard errors. Soil (S), RH vs. BK (RB) and their interaction (S × RB) were compared by two-way ANOVA, Duncan’s multiple range test (* *p* < 0.05; *** *p* < 0.001; ns: not significant). Different lowercase letters among bars indicate significant differences (*p* < 0.05).

**Figure 2 plants-13-00747-f002:**
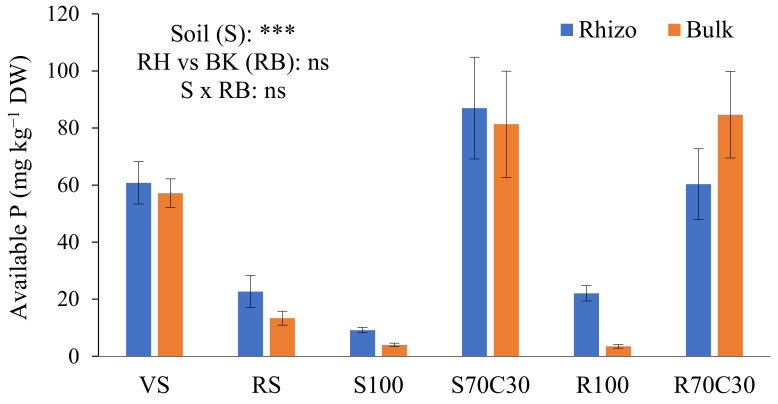
Concentration (mg kg^−1^ DW) of available P extracted by 0.5M NaHCO_3_ (buffered at pH 8.5, Olsen method) from volcanic soil (VS), red soil (RS), fluvial sand, pure (S100) and mixed with green compost (70:30 *v*:*v*; S70C30), and Mojave Mars regolith simulant MMS-1, pure (R100) and amended with green compost (70:30 *v*:*v*; R70C30), separated (after potato plant growth) in potato tuberosphere/rhizo (RH) and bulk (BK) soils. Bars indicate mean values ± standard errors (n = 5). Soil (S), RH vs. BK (RB) and their interaction (S × RB) were compared by two-way ANOVA, Duncan’s multiple range test (*** *p* < 0.001; ns: not significant).

**Figure 3 plants-13-00747-f003:**
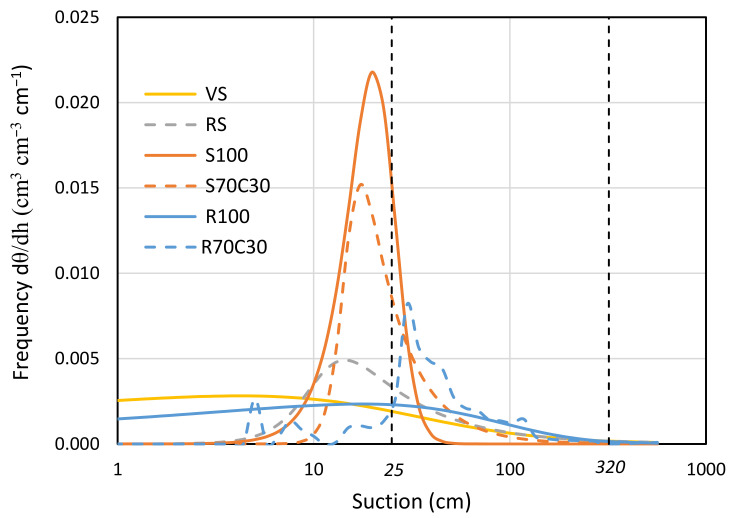
Pore frequency distribution of volcanic soil (VS), red soil (RS), fluvial sand, pure (S100) and mixed with green compost (70:30 *v*:*v*; S70C30), and Mojave Mars regolith simulant MMS-1, pure (R100) and amended with green compost (70:30 *v*:*v*; R70C30) with respect to suction, derived from first derivative of respective water retention curves.

**Table 1 plants-13-00747-t001:** Concentrations (g kg^−1^ DW) of organic C, total N and S, and C/N ratio, in volcanic soil (VS), red soil (RS), fluvial sand, pure (S100) and mixed with green compost (70:30 *v*:*v*; S70C30), and Mojave Mars regolith simulant MMS-1, pure (R100) and amended with green compost (70:30 *v*:*v*; R70C30), separated (after potato plant growth) in potato tuberosphere/rhizo (RH) and bulk (BK) soils. Data are shown as mean values ± standard errors (n = 5).

	C	N	C/N	S
	g kg^−1^ DW	g kg^−1^ DW		g kg^−1^ DW
VS	18.2 ± 0.9 b	1.53 ± 0.05 b	11.9 ± 0.29 b	0.83 ± 0.04 c
RS	15.2 ± 0.6 b	1.43 ± 0.06 b	10.6 ± 0.10 c	0.66 ± 0.04 c
S100	2.2 ± 0.2 c	0.18 ± 0.01 c	12.2 ± 0.34 b	0.09 ± 0.01 d
S70C30	72.8 ± 2.4 a	5.63 ± 0.26 a	13.0 ± 0.19 a	2.02 ± 0.09 b
R100	2.1 ± 0.2 c	0.30 ± 0.02 c	7.0 ± 0.31 d	0.13 ± 0.01 d
R70C30	75.4 ± 2.9 a	5.76 ± 0.22 a	13.1 ± 0.11 a	2.23 ± 0.10 a
Soil (S)	***	***	**	***
RH	31.6 ± 14.5	2.51 ± 1.10	11.6 ± 0.91	1.02 ± 0.40
BK	30.3 ± 14.3	2.43 ± 1.07	11.0 ± 1.04	0.96 ± 0.38
RH vs. BK (RB)	ns	ns	ns	ns
S × RB	ns	ns	ns	ns

Soil (S), RH vs. BK (RB) and their interaction (S × RB) were compared by two-way ANOVA, Duncan’s multiple range test (** *p* < 0.01; *** *p* < 0.001; ns: not significant). Different lowercase letters within each column indicate significant differences (*p* < 0.05).

**Table 2 plants-13-00747-t002:** Concentrations (mg kg^−1^ DW) of main macro- and micronutrients extracted by 1 M NH_4_NO_3_ from volcanic soil (VS), red soil (RS), fluvial sand, pure (S100) and mixed with green compost (70:30 *v*:*v*; S70C30), and Mojave Mars regolith simulant MMS-1, pure (R100) and amended with green compost (70:30 *v*:*v*; R70C30), separated (after potato plant growth) in potato tuberosphere/rhizo (RH) and bulk (BK) soils. Data are shown as mean values of 5 replicates.

	Ca	K	Mg	Na	Fe	Mn	Cu	Zn	B
	mg kg^−1^ DW								
VS	2895 b	1391 a	185 d	119 b	1.1 cd	0.40 c	0.27 a	0.51	0.08 c
RS	6425 a	260 e	282 b	108 b	1.8 b	0.66 b	0.05 d	0.24	0.01 c
S100	1060 d	65 f	72 e	28 c	1.7 bc	1.4 a	0.13 c	0.11	0.03 c
S70C30	1522 c	653 d	236 c	39 c	2.7 a	0.67 b	0.22 b	0.14	0.45 c
R100	2944 b	1069 b	510 a	286 a	0.8 d	0.30 c	0.13 c	0.08	6.7 b
R70C30	3140 b	947 c	505 a	308 a	0.7 d	0.28 c	0.11 c	0.01	8.6 a
Soil (S)	***	***	***	***	***	***	***	ns	***
RH	2979	801	318	137	1.5	0.59	0.17	0.18	2.0
BK	3016	661	279	159	1.4	0.65	0.14	0.17	3.3
RH vs. BK (RB)	ns	***	**	ns	ns	ns	ns	ns	***
S × RB	ns	***	**	*	ns	***	ns	ns	***

For the sake of clarity, this wide table shows only the mean values (n = 5), not followed by standard errors. Soil (S), RH vs. BK (RB) and their interaction (S × RB) were compared by two-way ANOVA, Duncan’s multiple range test (* *p* < 0.05; ** *p* < 0.01; *** *p* < 0.001; ns: not significant). Different lowercase letters within each column indicate significant differences (*p* < 0.05).

**Table 3 plants-13-00747-t003:** Concentrations (mg kg^−1^ DW) of main macro- and micronutrients extracted by 0.05 M EDTA (buffered at pH 7) from volcanic soil (VS), red soil (RS), fluvial sand, pure (S100) and mixed with green compost (70:30 *v*:*v*; S70C30), and Mojave Mars regolith simulant MMS-1, pure (R100) and amended with green compost (70:30 *v*:*v*; R70C30), separated (after potato plant growth) in potato tuberosphere/rhizo (RH) and bulk (BK) soils. Data are shown as mean values of 5 replicates.

	Ca	K	Mg	Na	Fe	Mn	Cu	Zn	B
	mg kg^−1^ DW								
VS	11,723 c	1046 a	437 b	98 b	109 d	43 c	19 a	14 a	0.8 b
RS	5204 e	215 e	286 c	85 b	291 a	610 a	14 b	6.3 b	0.7 b
S100	15,440 a	60 f	152 d	36 c	184 b	89 b	2.3 cd	1.8 c	0.5 b
S70C30	14,269 b	512 c	323 c	41 c	275 a	82 b	4.1 c	5.4 b	1.4 b
R100	5220 e	364 d	401 b	177 a	9.6 e	16 d	0.8 d	0.9 c	8.1 a
R70C30	8117 d	788 b	616 a	194 a	138 c	31 c	3.8 c	5.2 b	9.4 a
Soil (S)	***	***	***	***	***	***	***	***	***
RH	10,220	534	382	99	174	148	7.8	6.2	2.8
BK	9771	461	356	111	162	142	7.0	5.1	4.2
RH vs. BK (RB)	ns	***	ns	ns	ns	ns	ns	*	**
S × RB	ns	**	ns	*	**	ns	ns	ns	***

For the sake of clarity, this wide table shows only the mean values (n = 5), not followed by standard errors. Soil (S), RH vs. BK (RB) and their interaction (S × RB) were compared by two-way ANOVA, Duncan’s multiple range test (* *p* < 0.05; ** *p* < 0.01; *** *p* < 0.001; ns: not significant). Different lowercase letters within each column indicate significant differences (*p* < 0.05).

**Table 4 plants-13-00747-t004:** Concentrations (g kg^−1^ DW and mg kg^−1^ DW) of main nutrients in potato tubers cv. ‘Colomba’ grown on volcanic soil (VS), red soil (RS), fluvial sand, pure (S100) and mixed with green compost (70:30 *v*:*v*; S70C30), and Mojave Mars regolith simulant MMS-1, pure (R100) and amended with green compost (70:30 *v*:*v*; R70C30). Data are shown as mean values of 5 replicates.

	VS	RS	S100	S70C30	R100	R70C30	Sig.
g kg^−1^ DW							
C	419	399	419	417	419	420	ns
K	18.3 d	16.0 e	20.8 c	25.8 a	23.0 b	25.8 a	***
N	11.7 c	11.9 c	10.8 c	14.0 b	17.4 a	15.0 b	***
S	2.6 bc	2.4 c	3.1 ab	3.2 ab	2.9 abc	3.7 a	**
P	2.5 a	1.3 c	1.9 b	1.9 b	2.1 b	2.0 b	***
Mg	1.0 e	1.2 d	1.3 c	1.5 c	1.8 a	1.7 b	***
mg kg^−1^ DW							
Ca	197	419	179	366	265	357	ns
Na	37.8 c	66.4 bc	78.3 bc	88.2 abc	134 ab	144 a	**
Fe	23.2 ab	13.5 b	16.6 b	14.2 b	33.5 a	32.9 a	**
Zn	16.0 b	9.2 c	6.7 c	21.7 a	9.2 c	17.8 b	***
B	2.8 c	5.3 c	4.5 c	3.0 c	25.0 a	9.3 b	***
Mn	4.3 d	5.0 cd	5.6 bc	7.4 a	6.2 b	8.3 a	***
Cu	5.4 b	3.4 c	6.0 b	5.5 b	7.6 a	4.9 b	***

For the sake of clarity, this wide table shows only the mean values (n = 5), not followed by standard errors. Different letters within each row indicate significant differences according to one-way ANOVA, Duncan’s multiple range test (** *p* < 0.01; *** *p* < 0.001; ns: not significant).

**Table 5 plants-13-00747-t005:** Contents (g plant^−1^, mg plant^−1^ or µg plant^−1^ DW) of main nutrients in potato tubers cv. ‘Colomba’ grown on volcanic soil (VS), red soil (RS), fluvial sand, pure (S100) and mixed with green compost (70:30 *v*:*v*; S70C30), and Mojave Mars regolith simulant MMS-1, pure (R100) and amended with green compost (70:30 *v*:*v*; R70C30). Data are shown as mean values of 5 replicates.

	VS	RS	S100	S70C30	R100	R70C30	Sig.
g plant^−1^ DW							
C	7.4 ab	5.6 c	7.6 a	7.0 ab	3.4 d	6.5 bc	***
mg plant^−1^ DW							
K	322 c	227 d	376 b	436 a	189 d	401 ab	***
N	208 bc	169 d	198 c	237 a	141 e	233 ab	***
S	45.9 a	33.2 b	57.3 a	54.7 a	24.2 b	54.5 a	***
P	43.4 a	18.6 c	35.1 b	32.1 b	17.5 c	31.4 b	***
Mg	18.1 b	16.7 bc	24.4 a	24.5 a	14.7 c	25.8 a	***
Ca	3.4	5.8	3.3	6.1	2.2	5.6	ns
Na	0.7 b	0.9 b	1.4 ab	1.5 ab	1.1 b	2.2 a	*
µg plant^−1^ DW							
Fe	404 ab	188 c	302 bc	241 bc	277 bc	501 a	**
Zn	281 b	129 c	122 c	367 a	73 c	275 b	***
B	43.0 c	75.2 c	81.5 c	52.1 c	209 a	145 b	***
Mn	74.5 c	70.9 c	102 b	125 a	50.6 d	128 a	***
Cu	94.8 ab	48.8 d	108 a	92.5 ab	62.1 cd	75.4 bc	***

For the sake of clarity, this wide table shows only the mean values (n = 5), not followed by standard errors. Different letters within each row indicate significant differences according to one-way ANOVA, Duncan’s multiple range test (* *p* < 0.05; ** *p* < 0.01; *** *p* < 0.001; ns: not significant).

**Table 6 plants-13-00747-t006:** Quality of potato tubers cv. ‘Colomba’ grown on volcanic soil (VS), red soil (RS), fluvial sand, pure (S100) and mixed with green compost (70:30 *v*:*v*; S70C30), and Mojave Mars regolith simulant MMS-1, pure (R100) and amended with green compost (70:30 *v*:*v*; R70C30). Mean values ± standard errors; n = 5. Different letters within each row indicate significant differences according to one-way ANOVA, Duncan’s multiple range test (*p* < 0.05).

	Proteins	Starch	Total Dietary Fiber	Ascorbic Acid	α-Solanine	α-Chaconine	Total Glicoalkaloids
	(mg/g DM)	(%)	(g/100g dw)	(mg/100g fw)	(mg/kg dw)	(mg/kg dw)	(mg/kg dw)
VS	63.00 ± 0.46 d	54.52 ± 0.52 a	10.63 ± 0.05 b	18.77 ± 1.64	269.52 ± 15.39 ab	600.12 ± 39.59 a	869.64 ± 53.02 a
RS	45.64 ± 0.47 e	54.78 ± 0.18 a	9.34 ± 0.08 e	15.83 ± 0.58	214.76 ± 18.95 bc	427.69 ± 37.69 b	642.44 ± 50.13 b
S100	45.73 ± 0.26 e	48.43 ± 0.14 b	10.37 ± 0.09 d	17.24 ± 0.70	191.07 ± 22.93 c	395.25 ± 28.40 b	586.32 ± 44.44 b
S70C30	82.82 ± 0.61 a	47.49 ± 0.07 c	10.58 ± 0.07 bc	18.12 ± 0.56	234.80 ± 17.01 bc	484.49 ± 23.71 b	719.29 ± 40.40 b
R100	70.57 ± 0.26 c	48.17 ± 0.28 bc	11.17 ± 0.02 a	15.77 ± 0.95	320.57 ± 22.86 a	646.73 ± 27.38 a	967.29 ± 49.80 a
R70C30	75.68 ± 0.29 b	45.87 ± 0.36 d	10.42 ± 0.06 cd	16.86 ± 1.12	217.82 ± 16.63 bc	465.34 ± 30.35 b	683.15 ± 46.02 b

## Data Availability

The datasets of the experiments are available on request.

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
