# Peer review of "Assessment of Fertility Dynamics and Nutritional Quality of Potato Tubers in a Compost-Amended Mars Regolith Simulant"

_plants, 2024, doi:10.3390/plants13050747_

Round 1

Reviewer 1 Report

Comments and Suggestions for Authors

This study exemplifies an imperative step towards understanding the potential for growing food crops on Mars using simulated Martian regolith. The methodology described in this section provides a comprehensive overview of the physical, hydrological, and chemical analyses conducted on the substrates, as well as the assessment of plant growth and tuber yield, and the nutritional quality of potato. I have some points to discuss

1.      Regolith simulant MMS-1, may not fully capture the complexity of actual Martian regolith. Please comment.

2.      The conditions within the glasshouse may not fully mimic the environmental factors experienced on Mars, such as temperature variations, radiation levels, and atmospheric composition. Justify.

3.     Martian regolith simulant can provide similar ecosystem services to terrestrial soils, it's essential to recognize the inherent differences between Martian and Earth soils. Martian regolith may lack essential microbial communities and organic matter present in terrestrial soils, which could affect nutrient cycling and plant growth differently.

4.  Most importantly for such kind of experiments minimum two years data is required for better comparison.

5.     Please make all scientific names as italic.

6.      Several typographical errors are evident in the manuscript.

Comments on the Quality of English Language

minor changes required

Author Response

  1. Regolith simulant MMS-1, may not fully capture the complexity of actual Martian regolith. Please comment.

We agree with the reviewer, and we added the following sentence at lines 85-90 of the Introduction section: “Obviously, MMS-1 simulant does not represent the complexity of the entire surficial layer of Mars regolith, which shows high heterogeneity and spatial variability, analogously to Earth’s crust; nevertheless, its chemical composition and mineralogical patterns is as much as similar to those of Mars regolith collected by rovers and robotic spacecrafts.”

  1. The conditions within the glasshouse may not fully mimic the environmental factors experienced on Mars, such as temperature variations, radiation levels, and atmospheric composition. Justify.

We thank the reviewer for this comment. In principle, experiments on cultivation of higher plants for space application require specific facilities allowing to reproduce at best the environmental conditions of space vehicles and settlements (e.g., microgravity, cosmic radiation, altered atmospheric composition, climatic parameters). While some of these can be controlled in advanced growth chamber (i.e. temperature, relative humidity) or mimicked with relatively simple devises (e.g., microgravity in clinostats), some others need sophisticated and uncommon instruments (e.g., technologies to simulate ionizing radiation). Consequently, space research in plants is usually approached through a step procedure, and some experiments imply preliminary tests on large surface and higher number of plant samples compared to those possible in phytotrons and space module prototypes. Our team work on several aspects of plant cultivation in extraterrestrial environment. The experiment reported in the paper must be intended as a preparatory test for successive in-depth studies in facilities equipped for an advanced environmental control.

  1. Martian regolith simulant can provide similar ecosystem services to terrestrial soils, it's essential to recognize the inherent differences between Martian and Earth soils. Martian regolith may lack essential microbial communities and organic matter present in terrestrial soils, which could affect nutrient cycling and plant growth differently.

According to the reviewer’s statement, we inserted the following text at lines 393-398 of the Discussion: “Martian regolith and its terrestrial simulants, in fact, lack the essential microbial communities commonly present in the organic matter of the terrestrial soils, which have a key role in the rhizosphere nutrient cycling and plant growth processes [Duri et al. 2022]. Thus, when mixed with a quality compost, a Martian regolith simulant as MMS-1 can be able to provide similar ecosystem services to terrestrial soils.”

  1. Most importantly for such kind of experiments minimum two years data is required for better comparison.

We thank the reviewer for this comment. We agree that the result reproducibility is a crucial aspect of the research, and this is even more important for the studies implying living organisms, which interact with the surrounding environment. In this respect, as clarified in the response to the comment 2, our experiments are often carried out in consecutive steps with increasing level of definition. In the case of plant cultivation of potato plants on regolith-based substrates, the experiment represents a preliminary test and will be followed by the in-depth analysis of the crop response to the growing media in a last generation growth chamber, equipped for a complete control of environmental parameters, where growing cycles are replicated at least twice.

  1. Please make all scientific names as italic.

Done.

  1. Several typographical errors are evident in the manuscript.

We thank the reviewer for highlighting this weakness of the paper. The manuscript has been revised and typos and unclear sentences have been corrected.

During the technical check of your manuscript, we noticed that a high proportion of the cited references belong to you or your co-authors (Refs. 5,11,12,15,16,17,21,22,23,24,25,26,27). The self-citation rate of this manuscript is about 21.67%. According to our experience, it would be better if the self-citation rate of authors is less than 15% generally.

We deleted 4 of the 13 self-cited references. The self-citation rate of this manuscript now is 15%, as required.

In addition, we noticed that some of the content in the manuscript overlaps with your previous work. In order to avoid any potential cases, we recommend that you revise your manuscript appropriately. It would be best if the duplication rate of a single article is less than 10%. Please see the duplicate file attached.

We reduced as much as possible the duplication rate and overlaps with previous scientific works. However, only a few similarities were in the Results and Discussion while most of them were concentrated in the Materials and Methods section and in the captions of Tables and Figures. In this respect, changing the text would compromise the clarity and readability of the manuscript.

Reviewer 2 Report

Comments and Suggestions for Authors

The authors approached a topical subject as attention is given by the scientific community to exploring new worlds and exploiting their potential for human benefit. Not in the least bettering any kind of soil or growing substratum fertility properties is of great interest for food security worldwide.

The background presented in the Introduction is interesting and compelling and the aims of the research are clearly stated.

Data presentation is very detailed and clearly presented in tables and figures.

The Discussions give good and detailed explanations on data significance.

Materials and Methods are also detailed and in-depth presented. They are also appropriate for the research and its purpose.

The experiment whole description is very detailed and indicates much attention and thoroughness in unfolding the research.

The Conclusions briefly and correctly resume the research importance.

The research is complete with a rich list of fairly recent and appropriate references.

Author Response

We thank the reviewer for his comments. The manuscript has been further improved according to the requests of the other reviewer.

Round 2

Reviewer 1 Report

Comments and Suggestions for Authors

The ms has been improved and now can be accepted in its current state.